# Lives Saved in Low- and Middle-Income Countries by Road Safety Initiatives Funded by Bloomberg Philanthropies and Implemented by Their Partners between 2007–2018

**DOI:** 10.3390/ijerph182111185

**Published:** 2021-10-25

**Authors:** Delia Hendrie, Greg Lyle, Max Cameron

**Affiliations:** 1School of Population Health, Curtin University, Bentley, Perth 6102, Australia; greg.lyle@curtin.edu.au; 2Monash University Accident Research Centre (MUARC), Clayton, Melbourne 3800, Australia; max.cameron@monash.edu

**Keywords:** road safety measures, lives saved, traffic crashes, road fatalities, low- and middle-income countries

## Abstract

Over the past 12 years, Bloomberg Philanthropies (BP) and its partner organisations have implemented a global road safety program in low- and middle-income countries. The program was implemented to address the historically increasing number of road fatalities and the inadequate funding to reduce them. This study evaluates the performance of the program by estimating lives saved from road safety interventions implemented during the program period (2007–2018) through to 2030. We estimated that 311,758 lives will have been saved by 2030, with 97,148 lives saved up until 2018 when the evaluation was conducted and a further 214,608 lives projected to be saved if these changes are sustained until 2030. Legislative changes alone accounted for 75% of lives saved. Concurrent activities related to reducing drink driving, implementing legislative changes, and social marketing campaigns run in conjunction with police enforcement and other road safety activities accounted for 57% of the total estimated lives saved. Saving 311,758 lives with funding of USD $259 million indicates a cost-effectiveness ratio of USD $831 per life saved. The potential health gains achieved through the number of lives saved from the road safety initiatives funded by Bloomberg Philanthropies represent a considerable return on investment. This study demonstrates the extent to which successful, cost-effective road safety initiatives can reduce road fatalities in low- and middle-income countries.

## 1. Introduction

Based on the 2016 WHO Global Health Estimates, the number of global road traffic deaths continues to increase annually from 1.15 million in 2000 to 1.35 million in 2016 [1]. This burden is disproportionately borne by those living in low- and middle-income countries (LMICs), with death rates in low-income countries three times as high as rates in high-income countries [1]. To address this persistent problem, the Decade of Action for Road Safety was proclaimed in 2010 [2] in order to support the development of national and local actions, with countries encouraged to implement activities according to a five-pillar action plan: road safety management, safer roads and mobility, safer vehicles, safer road users, and post-crash response [2,3]. Further highlighting the need for action, the United Nation Sustainable Development Goals include two road safety targets: Target 3.6, which aims to reduce global road traffic deaths and injuries by 50% by 2020, and Target 11.2, which aims to provide access to safe, affordable, accessible, and sustainable transport systems for all by 2030 [4].

The scale of the problem is large, and the funding needed to address it, especially in LMICs, has in the past been inadequate [3]. To help address this lack of resources, numerous international, national, and local programs have been established, and extensive literature has been published on their effectiveness for saving lives. In this paper, we focus on the initiatives funded by Bloomberg Philanthropies (BP) and implemented by their partner organisations (Appendix A) within three phases of the Bloomberg Road Safety Program. Initially, a $9-million pilot program was undertaken between 2007 and 2009, the Bloomberg Pilot Road Safety Program (Pilot Phase (2007–2009)). This program funded the publication of a Global Road Safety Status Report [5], with additional reports released over the next decade to monitor the progress of countries worldwide. The program also funded the implementation of country-based demonstration projects with the purpose to highlight what could be achieved through cost-effective road safety interventions [6], which have proven to be effective in high-income countries [7,8]. A significant activity in this program was the introduction of the national mandatory helmet legislation in Vietnam [9]. In 2010, the pilot was expanded, with BP committing $125 million over five years to the Global Road Safety Program (GRSP (2010–2014)) [10]. This funding targeted 10 LMICs (known initially as the Road Safety in 10 countries (RS-10) project [11]), which accounted for almost half of global road traffic-related deaths [10]. The remit of this commitment was to (1) strengthen road safety legislation; (2) implement life-saving interventions that increased seatbelt, child-restraint, and helmet use and reduced speeding and drunk driving; (3) survey road networks and recommend infrastructure improvements; (4) build sustainable transport alternatives; and (5) improve the collection of data to evaluate project effectiveness and better target interventions [10,11]. Extensive effort during this time was undertaken to create baseline datasets for local road crash fatalities and risk factors in Brazil [12,13], Cambodia [14,15,16,17], China [18,19,20], Egypt [21,22,23], India [24,25,26,27,28], Kenya [29,30,31,32], Mexico [23,33,34], Russia [23,35,36], Turkey [23,37,38], and Vietnam [39,40,41,42]. These datasets have fed into subsequent global status reports [1,43]. However, data availability and quality issues still exist [3]. In addition to the GRSP (2010–2014) funding of interventions to promote road safety and facilitate changes in road safety legislation, BP also funded investigations into the road safety impacts of high-performance bus rapid-transit systems and busway design features in nine Bus Rapid Transit (BRT) systems around the world [44]. These interventions complemented road infrastructure improvements by the World Bank and World Resources Institute/EMBARQ in other selected countries [45].

In order to assess the benefits of these interventions, a range of observational site-based evaluations have also been funded under the program. Evaluation has shown reductions in risk factors in speeding and increased seatbelt and child-restraint use in regions of Russia [36,46]; an increase in helmet use in provinces of Viet Nam [42]; and an increase in seatbelt use but uneven improvements in speeding in the provinces of Turkey [47]. Evaluations at the city level have shown a decrease in the prevalence of drink driving and speeding in China [19]. Other studies focusing on the reduction in fatalities as an outcome measure have shown reductions in crashes but not deaths from a range of nationally based interventions in Mexico [48] and reductions in fatalities from the construction of a bus rapid-transport system in India [49]. More comprehensive evaluations of the potential lives saved have also been undertaken to elucidate the benefits of the whole programme of interventions. In 2012, 10,310 lives were initially projected to be saved from the implementation of 12 road safety interventions within 10 LMICs from 2011 to 2015 under the GRSP (2010–2014) [50]. Further analysis of the law and regulation changes in six LMICs under the program reported an estimated 19,000 lives saved from 2007 to 2013 and 90,000 lives to be saved from 2014 to 2023 [51,52].

A second phase of the global program was launched in 2015, the Bloomberg Philanthropies Initiative for Global Road Safety (BIGRS (2015–2019)). The initiative invested a further $125 million over five years to five LMICs and 10 cities with a high burden of fatalities and injuries, and an investment in improving vehicle safety was added to the program [53]. More baseline data was collected on fatalities and risk factors in Bangladesh [54]. Directed by past experiences, BP narrowed its focus on interventions into three key areas: (1) strengthening national legislation, (2) implementing road safety interventions at the municipal level, and (3) promoting vehicle safety. Observational studies have showed decreases in speeding in Bandung, Indonesia, Bangkok, Thailand, Ho Chi Minh City, Vietnam [55], and Sao Paulo, Brazil [56]. This later result also saw a decrease in drunk driving and a continuing high use of seatbelt and helmets. Additionally, an increase in speed enforcement was estimated to produce a four-percent decline in the total number of fatalities per year on roads in Bogota, Brazil [57].

Since 2007, a large numbers of road safety interventions have been implemented by BP and partner organisations, many of which have not been evaluated to understand their significance within the LMICs. Therefore, the aim of this study was to estimate the potential lives saved by road safety initiatives supported by BP and its partner organizations in the pilot, first, and second phases of the program (2007 to 2018) and to project future reductions in mortality through to 2030.

## 2. Materials and Methods

### 2.1. The Interventions Evaluated and Evalauation Workflow

The list of interventions evaluated in the pilot and two phases of the program are shown in Figure 1. The evaluation time period was to the end of the year 2018 in order for findings to be used in decisions about future funding following completion of BIGRS (2015–2019). The evaluation re-evaluated initiatives that were initially evaluated to the year 2023 for the Pilot Phase (2007–2009) and GRSP (2010–2014) [52]) as well as initiatives that were not evaluated in these phases [58,59] and initiatives in the BIGRS (2015–2019).

Figure 2 provides an overview of the data workflow for the evaluation of each type of road safety initiative.

### 2.2. Road Safety Legislation—Country-Based Initiatives

Changes in national road safety legislation were implemented in the Pilot Phase (2007–2009), the GRSP (2010–2014), and the BIGRS (2015–2019).

Annual national fatality data were extracted from the World Health Organization (WHO) website [60] for the years 2000, 2010, 2015, and 2016. The number of fatalities for the missing years in this time interval was estimated based on the annual linear increase between known fatalities. Adjustments were made to the linear trend calculation based on published fatality data from the Global Health Data Exchange [61] for the same time period. Annual fatalities to 2030 were extrapolated from the average annual linear increase between 2010 and 2016. To address specific legislation changes for specific road-user groups, road-user group fatalities were estimated by multiplying total fatalities by the ratio of country-specific road-user group fatalities from published observations for the same time period [61]. This method produced estimates for total and road-user specific road fatalities from 2000 to 2030 for each country where legislation changes were implemented.

The number of lives saved for the 2000 to 2030 time period was calculated by multiplying estimated lives lost without the intervention times the percentage reduction in deaths attributable to the intervention [52]. The method used is below, under the assumptions that the future estimated fatalities and the effectiveness ratios continued to the year 2030.
LS_LR_ = IF_PRE-LR_ × ΔP_RF_ × EFF_LR_
where LS_LR_ is the lives saved due to the law, IF_PRE-LR_ is the number of fatalities without the law, ΔP_RF_ is the change in prevalence of the risk factor behaviour, and EFF_LR_ is the effectiveness of the law in reducing fatalities with adjustments made to account for the level of police enforcement.

### 2.3. Social Marketing Campaigns, Police Enforcement, and Related Road Safety Activities

#### 2.3.1. Data to Estimate the Lives Saved from GRSP (2010–2014) Sub-National Initiatives

Social marketing campaigns, police enforcement, and related road safety activities of the GRSP (2010–2014) were implemented at the sub-national level, i.e., regions, provinces or oblasts within 10 countries [58]. Due to the lack of an archive of published fatality data at the sub-national areas, a population-based fatality rate method was used to construct current and future annual projections of deaths in these areas.

Population data for the sub-national areas were taken from online or published sub-national statistics. Population counts were interpolated for years with no population data based on the annual linear increase between the years with a known population. Populations were extrapolated forward as an annual linear increase to 2030. This was based on the average annual population change between years with known populations, usually calculated as the five-year average between the most recent years.

Annual fatality data or fatality rates for the sub-national areas were taken from Bloomberg Philanthropies Global Road Safety Program Final Evaluation Reports, other publications [62], and extracted from online sub-national statistics. However, only a small range of years of road fatality data was available for these sub-national areas. Where annual fatality data were available, annual fatality rates were calculated from observed fatalities and corresponding annual population. Fatalities for missing years was estimated by multiplying the annual population by an average of fatality rates for nearby years. Annual fatalities to 2030 was extrapolated from annual population and the five-year average of the most recent known death rates. Road-user group fatalities were calculated by multiplying total fatalities by the ratio of fatalities by road-user group for the area if available or for the area’s corresponding country [61].

#### 2.3.2. Data to Estimate the Lives Saved from BIGRS (2015–2019) City Initiatives

Social marketing campaigns, police enforcement, and related road safety activities of BIGRS (2015–2019) were implemented at the city level. Due to the lack of an archive of published fatality data at the city level, a population-based fatality rate method was used to construct current and future annual projections of deaths in these cities.

City populations were taken from published online estimates and were interpolated for years with no population data based on the annual linear increase between the years with a known population. Populations were extrapolated forward as an annual linear increase to 2030 based on the average annual population change between years with known populations, usually calculated as the five-year average between the most recent years.

Annual fatality data were taken from Bloomberg Philanthropies Initiative for Global Road Safety reports or calculated from published rates. Where annual fatality data were available, annual fatality rates were calculated from observed fatalities and corresponding annual population. Fatalities for missing years were estimated by multiplying the annual population by an average of fatality rates for nearby years. Where no fatality data were available for a city, fatality rates published at the corresponding regional or country level were used. Annual fatalities to 2030 were extrapolated from annual population and the five-year average of the most recent known death rates. Road-user group fatalities were calculated by multiplying total fatalities by the ratio of fatalities by road-user group for the area if available or for the area’s corresponding country [61].

#### 2.3.3. Method to Estimated Lives Saved

Estimated lives saved was calculated based on the changes in behavioural risk factors caused by the initiative. The change in risk factor was measured as the difference between the population attributable risk fraction [63,64] calculated before and after the initiative. The fraction calculation is based on length of time the initiative was undertaken, the relative risks on mortality (Appendix B), and the observed changes in annual prevalence reported by observational data at each intervention site. For these initiates, observational data were collected by the John Hopkins International Injury Research Unit for each initiative evaluated. The formula used for the population attributable risk fraction is shown below:$$\left[P_{\mathrm{RF}}\;\times \;(\mathrm{RR}-1)]/[P_{\mathrm{RF}}\;\times \;(\mathrm{RR}-1)\;+\;1\right]$$
where P_RF_ is the prevalence of the behavioural risk factor in the population, and RR is the relative risk of the probability of dying among the population exposed to the behavioural risk factor compared with the unexposed population. Note that in cities where speeding prevalence was available by level, the population attributable risk fraction was calculated by the formula for polytomous risk categories [65].

Estimated annual lives saved was calculated by the change in population attributable risk fraction due to the initiative, multiplied by the annual fatalities for 2011 to 2018. Annual lives saved from 2019 to 2030 were estimated by multiplying the number of fatalities by the average reduction in population attributable risk fractions. This assumed that the effectiveness of the behavioural intervention continued at the same level of campaign intensity and/or enforcement to 2030.

### 2.4. GRSP (2010–2014) Safety Impact of Sustainable Urban Transport Alternatives

Lives saved for bus rapid-transit projects and urban development projects in four countries, policy- and capacity-building projects in three countries, and black spot identification and recommendations and bicycle safety projects in one country were estimated within the report by World Resources Institute/EMBARQ [59]. Estimates of lives saved between 2010 and 2014 and projected to 2019 were based on the difference between the baseline scenario using actual fatal crash counts before implementation and the post-implementation scenario based on the either actual or expected fatal crash counts based on estimates from the literature evaluating the safety impact of these initiatives elsewhere. In policy or engineering projects, lives saved were calculated as the reduction in fatalities, the difference pre- and post-initiative from changes in vehicle travel, exposure to traffic crashes, speed reductions, and traffic calming. Lives saved from bicycle safety projects were estimated by taking into account the growth in the number of cyclists as a result of the cycling projects and the estimated reduction in crash risk.

Projections of lives saved were made by expanding the World Resources Institute/EMBARQ timeframes, originally estimated to 2019, to 2030 using the same World Resources Institute/EMBARQ method (Empirical Bayes Method) to estimate crash trends under each scenario. When more recent ex-ante site crash data were available beyond the 2015 report [59], the project scenarios were updated so that estimated lives saved for the time period between 2014 and 2018 could be recalculated. To estimate baseline and project scenario crash, injury, and fatality counts beyond 2018, the weighted average of observed trends in site and city-wide injury and fatality trends were updated for the years between 2014 and 2018 when the data were available.

### 2.5. Road Infrastructure Improvements

#### 2.5.1. BIGRS (2015–2019) Road Infrastructure Improvements (World Resources Institute/EMBARQ))

Estimates of the expected impact on lives saved in 10 target cities from the implementation of road infrastructure, speed calming measures that included setting lower speed limits in selected urban areas, and sustainable transport infrastructure improvements were provided by World Resources Institute/EMBARQ for 2015–2034 [62].

A variety of methods were applied to estimate the number of lives saved, and their selection depended on the availability, quality, and completeness of the data collected. The most common method used to estimate lives saved was the crash reduction fraction (CRF). The CRF represents the effectiveness of a given type of road safety infrastructure improvement, and a range of values were obtained through literature reviews of the effectiveness of the interventions implemented elsewhere. Selection of the appropriate CRFs for the evaluation of each project was then based on a range of expert opinions on the value’s validity and applicability to the environment of the project sites.

The number of lives saved was calculated by multiplying the fatalities recorded at a specific site over a number of years before the project was implemented by the CRF and the time frame over which lives saved are calculated. The lives saved due to each of these improvements were calculated as:
LS_IN_ = SF/y × CRF × Y_n_
where LS_IN_ is the lives saved due to infrastructure improvement, SF/y is the site fatalities per year over a number of years before the project, CRF is the crash reduction fraction/100, and Y_n_ is the timeframe over which the lives saved are calculated.

Projections of lives saved were made by multiplying the estimated number of lives saved for each project by the number of years between the year the project was implemented and the year 2030.

#### 2.5.2. BIGRS (2015–2019) Road Safety Risk Assessment and Road Infrastructure Improvements by the World Bank Global Road Safety Facility

Estimated lives saved for road infrastructure improvements in five target countries and 10 target cities was calculated through the International Road Assessment Programme (iRAP) Star Rating Methodology [66]. The Star Rating methodology involves a video-based assessment of over 50 road attributes that specify the safety risk associated with vehicle occupants, motorcyclists, bicyclists, and pedestrians. These safety risks are then used to compute the relative risk of death on a 100-m segment of the road. Based on this risk analysis, the iRAP methodology further proposes a list of all possible lifesaving recommendations (a Safe Road Investment Plan). These will estimate a net positive return on investment and estimate the annual number of lives saved due to the interventions compared with the existing configurations. Lives saved were estimated to evaluate improvements on 2185 km of road network where each of the countries have incorporated the recommendations for road infrastructure improvement. The lives saved metric was estimated cumulatively, from 2015 until 2018, and then further extrapolated to 2030.

## 3. Results

The contribution to the number of lives saved was determined by the specific initiative undertaken in the program and the location in which they were implemented.

### 3.1. Total Estimated Lives Saved Attributed to the Bloomberg Road Safety Program

From 2007 to 2030, the total number of lives saved was estimated at 311,758 lives across all initiatives implemented by the program (Table 1). The introduction of the Pilot Phase (2007–2009) and the two phases (GRSP (2010–2014) and BIGRS (2015–2019)) of the program have been estimated to have saved 97,148 lives up until the end of 2018, when the evaluation was conducted. A further 214,608 lives are projected to be saved if these changes are sustained until 2030. Initiatives implemented between 2007 and 2014 (Pilot Phase (2007–2009) and the GRSP (2010–2014)) have had the greatest contribution, with a total of 288,887 or 93% of lives saved to 2030. This number reflects the establishment of these initiatives in the time period to 2014 in a large number of countries, some with high population densities.

### 3.2. Estimated Lives Saved Due to Legislation Changes

Legislation changes to national road safety legislation impacted nearly 3.5 billion people in ten countries over the whole program. These initiatives accounted for 234,458 lives or 75% of total lives saved from all initiatives (Table 1). Changes in laws in the countries of Vietnam and China made up 34% of total lives saved by legislative changes, followed by Thailand (14%), Turkey (13%), Kenya (7%), and India (4%) (Appendix C).

The number of lives saved differed by the focus of the road safety initiative. Initiatives that led to reductions in impaired driving in seven countries made up 64% of the total number of lives saved by legislative changes. The majority (88%) were from changes to legislation in China and Vietnam (Appendix C). Changes to legislation in motorcyclist protection in five countries and speed management in two countries made up a further 18% and 14% of lives saved, respectively.

### 3.3. Estimated Lives Saved Due to Social Marketing Campaigns, Police Enforcement, and Related Road Safety Activities

The total lives saved from social marketing campaigns, police enforcement, and related road safety activities was estimated at 40,919 lives to 2030. The lives saved from the two phases of the program between 2010 and 2018 have been estimated at 9093 lives. If these activities continue to 2030, a further 31,826 lives are estimated to be saved.

The road safety initiatives conducted within the GRSP (2010–2014) in subnational areas of 10 countries contributed to 79% of the total lives saved from social marketing campaigns, police enforcement, and related road safety activities until 2030. The biggest contributors were activities that reduced levels of impaired driving (78%) and speeding (18%) (Table 1). Over half of lives saved across all activities within the GRSP (2010–2014) were from initiatives targeting impaired driving in the Hyderabad area of India (Appendix D).

Road safety activities across the target cities within BIGRS (2015–2019) were projected to save 8453 lives to 2030. This is largely from a reduction in speeding (48%) and impaired driving (45%) (Table 1). The majority of these lives saved due to the introduction and continuation of risk factor focus initiatives were in the cities of Sao Paulo, Brazil (39%); Bogota; Colombia (23%); and Addis Ababa, Ethiopia (22%) (Appendix E).

### 3.4. Estimated Lives Saved Due to the Safety Impact of Sustainable Urban Transport Alternatives

Improvements within the GRSP (2010–2014) focused on making roads safer by reducing the need for car travel and encouraging the use of other transport modes. The introduction of these initiatives implemented by World Resources Institute/EMBARQ have been estimated to have saved 8156 lives until 2018 (Table 1). Total lives saved through the continuation of these initiatives through to 2030 is projected to be 27,363 lives saved. Among the four countries in which these initiatives were implemented, activities in Brazil (39%) and Mexico (34%) made the greatest contribution to the lives saved (Appendix F).

### 3.5. Estimated Lives Saved Due to Road Infrastructure Improvements

Initiatives implemented in 10 cities under the guidance of the World Resources Institute/EMBARQ in the BIGRS (2015–2019) were projected to save 3907 lives until 2030. Almost all (90%) were attributed to the projected lives saved for the 2019 to 2030 time period rather than for the four years (2015–2018) since their implementation. The majority of lives saved were improvements in the cities of Bogota, Colombia (23%) and Fortaleza, Brazil (19%) and Ho Chi Minh City, Vietnam (16%) (Appendix G).

A total of 5111 lives were estimated to be saved from road infrastructure improvements under the guidance of the World Bank Global Road Safety Facility. Similar to the estimates under the World Resources Institute/EMBARQ initiatives above, 82% of lives saved were projected between the years 2019 to 2030. A total of 4596 lives were estimated to be saved within five countries, the majority in India (68%). The remaining lives saved were from city interventions in Fortaleza, Brazil and Ho Chi Minh City, Vietnam, which made up over half the lives saved in these city-based road improvements (Appendix H).

## 4. Discussion

The evaluation of the pilot and two phases of the road safety initiatives undertaken by Bloomberg Philanthropies and partner organizations has been estimated to save a total of 311,758 lives over the 2007 to 2030 time period. The program of work was estimated to cost USD $259 million, which indicates a cost-effectiveness ratio of approximately USD $831 per life saved. The cost-effectiveness ratio is widely used to measure the costs and consequences of two or more alternatives, in this case, the Bloomberg-funded initiatives versus the status quo or “do nothing” option. To determine whether this represents a good return on investment, we can compare this value to the statistical value of life estimate, which is an estimate of the value society places on reducing the risk of dying. This “rule of thumb” calculation developed by the International Road Assessment Programme (iRAP) is used to assess value for money in LMICs and is calculated by multiplying the gross domestic product (GDP) per capita of a country by 70 [66]. This metric has been proven to provide a good indicator of the relevant aspects needed to guide its use as a guiding principle for assessing road interventions. Additionally, it can be used to quickly estimate the return of any road safety intervention, especially in LMICs, where resources are limited, and the efficient allocation of resources is fundamental to maximising deaths averted [67]. For the countries examined here, Kenya reported the lowest statistical value of life, with a value of USD $132,300 based on a GDP per capita of USD $1890 in 2018 [68]. Other countries where initiatives have been undertaken, such as India (USD $2100), Vietnam (USD $3420), and Russia (USD $11,510) have higher GDP per capita values and therefore higher statistical values of life, at USD $147,000, USD $239,400, and USD $805,700, respectively. These estimates are between 169 and 970 times higher than the estimated average cost per life saved of USD $831 spent by BP. These returns demonstrate a considerable return on investment from the implemented road safety initiatives.

This evaluation demonstrated the variation in the effectiveness of initiatives in terms of the lives saved and can inform a degree of targeting of future investment. The tightening of weak legislation offers a very simple tool to save lives. We found this in the area of impaired driving, which was by far the most effective measure to reduce fatalities. This result reiterated the conclusions by an earlier evaluation of the Bloomberg Road Safety Program from 2008 to 2023 [52], which showed that much of the gain in lives saved was due to changes in seatbelt and motorcycle helmet legislation. Changes to legislation offer a wide reach to change road safety behaviours (i.e., speed management or impaired driving) across large populations. Once implemented, these changes cannot easily be repealed, and while some relaxations of the laws exist [69], they still ensure that these benefits of lives saved are maintained into the future to some degree. Additionally, changes to legislation are a highly cost-effective tool from a government perspective given that the costs of passing and implementing these legislative changes are likely to be minimal [70].

Changes to legislation offers a solid lever for reducing fatalities, but the act of changing the legislation on its own without additional activities can have minimal effect on changes in behaviour. For example, the prevalence of helmet wearing in Kenya remained similar after the passage of a traffic amendment bill [30]. The approach of supporting these legislative changes with social media and public education campaigns and enhanced enforcement provides additional benefits in terms of lives saved. Using a multi-pronged, coordinated, and long-term approach [16] increases the populations’ understanding of why the new law has been introduced and, through experiencing local-level campaigns of police enforcement of these new laws, a perception that these laws will be continually enforced. We found that, regarding lives saved, the multi-pronged approach of interventions to reduce drunk driving, both legislative changes and social marketing campaigns run in conjunction with police enforcement and related road safety activities, accounted for a significant share (57%) of the total estimated lives saved. This approach takes into account the conclusions from evidence-based research [16]. This study showed through a roadside survey on drunk driving in Cambodia that respondents did not often comprehend what amount of intake constitutes the legal limit in terms of number of drinks and the perception that their driving was impaired when drinking alcohol. Furthermore, while being stopped by the police was a deterring factor against drunk driving, only a small number of respondents had been pulled over.

Improvements to road infrastructure offer a long-term structural intervention to save lives. These types of interventions are not easily undone, and therefore, they are beneficial in saving lives into the future. Spending on road maintenance and conservation has a positive impact on road safety, with costings from European countries suggesting that the death rate would be reduced by 0.025 for each thousand euros invested [71]. Spending on maintenance is also much more relevant to road safety than spending on construction, and governments should be encouraged to implement maintenance programs for efficient connections but also to enhance safety standards [72]. Further studies are needed to provide further evidence of the benefits of these types of interventions in LMICs.

### Limitations

The estimated effectiveness of road safety interventions measured by lives saved is a function of the number of road crash fatalities pre-intervention over time, exposure to the risk factor being addressed, and the effectiveness of the intervention in reducing fatal crashes. Additionally, if an intervention targets a specific risk factor and/or road-user group, data are required about the number of road crash fatalities attributable to that risk factor and/or road-user group. The quality of data collected [60] and representativeness of the estimates used in the calculations for each LMIC limit our evaluation and final estimate of lives saved.

Reported data on road fatalities were available for some countries, regions, and cities; however, there is consistent under-reporting of road fatalities in LMICs, with uncertainty in estimates of fatalities larger than those obtained for injuries and crashes [73]. Where reported road crash deaths were not available, the analysis relied on the use of modelled estimates of fatalities from global data sets. These estimates a have degree of uncertainty attached to them as an evidence base for impact assessment [50]. For example, the WHO death data used for country analysis had four levels of quality: high for Brazil and Mexico; medium for Philippines, Russia, and Turkey; and low and very low for Cambodia, China, India, Kenya, Thailand, Vietnam, and Tanzania [60]. Where mortality breakdowns by road-user group were required, estimates were modelled from the Global Burden of Disease study [61]. The road crash fatality data in target cities and subnational areas had additional limitations, with data not consistently available for all years. In years with no data, road fatalities were estimated based on past trends and where data were available, in fatalities or fatality rates and populations. The reliability of these national, regional, and city populations estimates ranges from fairly robust estimated provided by government censuses to rubbery estimates of populations published on internet websites. Estimates of populations into the future were largely based on past projections, and these may not represent future growth of the population. In the case of road improvement initiatives, road crash data were not always available, so estimates from similar networks were used. The projections of future mortality were based on an extrapolation of past trends, i.e., the assumption that initiatives are sustained, and their gains are maintained into the future, and as such, a past trend may not mirror future trends. Additionally, in projecting to 2030, no account was taken of likely growth in traffic over the period, so the estimated lives saved are likely to be an underestimate.

Observational studies provided useful information on exposure to risk factors. These studies assume changes in prevalence are attributed to the interventions, which may overstate the effectiveness of interventions given the presence of other ongoing road safety activities. When these data were not available, we were reliant on the best available data, often from high-income countries rather than from the low- and middle-income countries themselves. Effectiveness of interventions is also dependent on the extent of law enforcement, and while we downward-adjusted these estimates based on local knowledge, enforcement is likely to be less in the countries investigated. The estimated lives saved by road improvements may also have been overestimated, as the magnitude of life saved was evaluated outside of the effect of other road safety measures that could be implemented concurrently [72]. The assumptions that we were forced to make increases the uncertainty around the estimates of lives saved.

Limited research has been performed investigating the combined effects of road safety measures and the extent to which the effect of a given safety measure is impacted when another safety measure is introduced [74]. We adopted the method that is commonly used, namely the method of common residuals, which assumes the effects of road safety measures are independent of one another [75]. If this assumption is too optimistic, the effectiveness of interventions in saving lives will be an overestimate.

The time frame in which the evaluation was conducted, till the end of 2018, will underestimate the effectiveness of the BIGRS 2015–2019 program because it does not fully include all 2019 initiatives. For example, several policy successes from the 2015–2019 phase were not counted in this evaluation because they were passed after December 2018, including the Motor Vehicles Amendment Act in India, the Child Restraint Safety Act in the Philippines, and the strengthening of several vehicle safety standards [76,77]. Additionally, the World Bank and WRI reported after the 2018 cut-off date that several of their infrastructure recommendations have now been incorporated by governments and were not included. It is also important to note that currently, we do not have a scientific approach to measure the impact of several city-based interventions, such as updating design codes; however, it is believed that they do have a life-saving impact.

Our calculations also do not specifically acknowledge the contribution of many partner organizations working with BP to reduce the burden of road fatalities in the target countries and cities. The work of these partner organizations is central to achieving this collective goal despite their activities not always being able to be directly measured in tangible outcomes, such as lives saved. Successful implementation of initiatives requires a multi-sectoral approach and a broad base of support from government agencies, regional administrators, and civil society organizations, which needs to be fully acknowledged in achieving the outcomes of the road safety initiatives.

## 5. Conclusions

In this manuscript, we have reported the extent to which 12 years of road safety campaigns funded by Bloomberg Philanthropies and implemented by their partner organisations can contribute to saving a considerable number of lives while providing a high potential return on investment. This evidence acts as justification for future investment in road safety programs in low- and middle-income countries. This is needed given the expected increase in traffic exposure due to rapid motorisation of transport; the persisting weaknesses in road safety standards, vehicle safety, and maintenance; and in the design and implementation of policies and safe transportation infrastructure [73].

## Figures and Tables

**Figure 1 ijerph-18-11185-f001:**
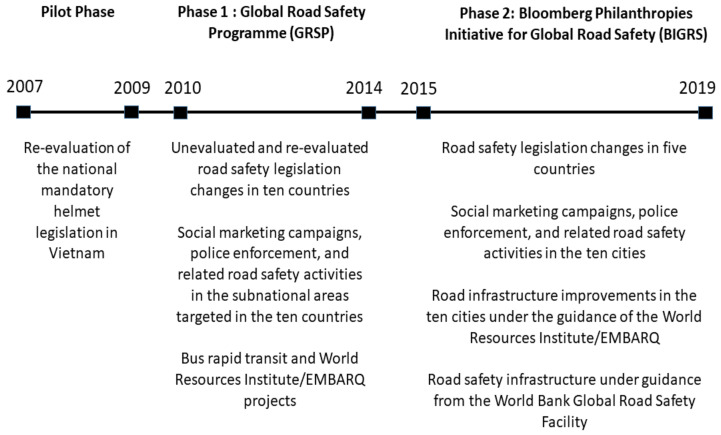
The evaluations undertaken under the Pilot Phase (2007–2009) and the GRSP (2010–2014) and the BIGRS (2015–2019) phases of the Bloomberg Road Safety Program.

**Figure 2 ijerph-18-11185-f002:**
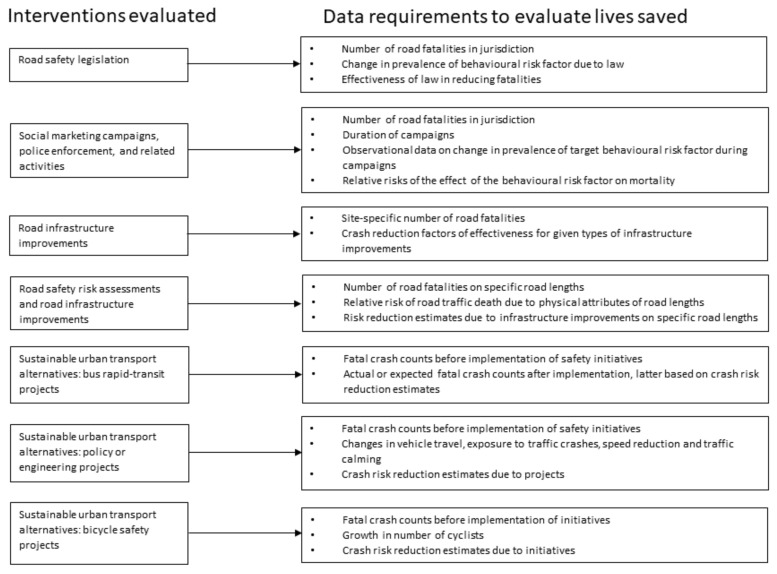
Data requirements to evaluate lives saved by intervention type.

**Table 1 ijerph-18-11185-t001:** Summary of lives saved by road safety initiatives undertaken by Bloomberg Philanthropies and partner organizations, 2007–2030.

Road Safety Initiative	Number of Lives Saved by Road Safety Interventions		
	Until 2018 (*n*)	2019–2030 (*n*)	TOTAL (*n*)
**Legislative changes** ** *Pilot Phase and GRSP (2007–2014)* **			
Impaired Driving	53,090	96,024	149,114
Motorcycle Protection	12,739	28,061	40,799
Safety Belts	1987	5163	7150
Speed Management	10,168	21,827	31,995
**Subtotal**	**77,984**	**151,075**	**229,058**
** *BIGRS (2015–2019)* **			
Impaired Driving	63	494	557
Motorcycle Protection	119	1142	1261
Safety Belts	446 ^2^	3135 ^2^	3581
**Subtotal**	**628**	**4771**	**5399**
**Total**	**78,612 ^1^**	**155,845**	**234,458**
**Social marketing campaigns, police enforcement, and related road safety activities**			
** *GRSP (2010–2014)* **			
Impaired Driving	5552	19,707	25,259
Motorcycle Protection	41	91	132
Safety Belts	463	824	1287
Speed Management	1949	3838	5787
**Subtotal**	**8006**	**24,460**	**32,466**
** *BIGRS (2015–2019)* **			
Impaired Driving	516	3259	3775
Motorcycle Protection	43	282	325
Safety Belts	55	233	288
Speed Management	474	3592	4066
**Subtotal**	**1087**	**7366**	**8453**
**Total**	**9093**	**31,826**	**40,919**
**Sustainable urban transport alternatives**			
**GRSP (2010–2014)**			
World Resources Institute/EMBARQ Safety Impact of sustainable urban transport alternatives	**8156**	**19,206**	**27,363**
**Road infrastructure improvements**			
**BIGRS (2015–2019)**			
World Resources Institute/EMBARQ Road infrastructure improvements	389	3518	3907
iRAP assessments of road safety infrastructure and subsequent incorporation of recommendations	898	4213	5111
**Total**	**1287**	**7731**	**9018**
**Pilot Phase and GRSP (2007–2014) Total**	**94,146**	**194,741**	**288,887**
**BIGRS (2015–2019) Total**	**3002**	**19,868**	**22,870**
**TOTAL**	**97,148 ^1^**	**214,608**	**311,758**

^1^ Includes 2586 lives saved from Vietnam helmet-wearing law during the Pilot Phase 2007–2009); ^2^ includes lives saved estimates for safety belts in China, see Appendix C.

## Data Availability

Not available.

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
