# Peer review of "Lives Saved in Low- and Middle-Income Countries by Road Safety Initiatives Funded by Bloomberg Philanthropies and Implemented by Their Partners between 2007–2018"

_ijerph, 2021, doi:10.3390/ijerph182111185_

Round 1
Reviewer 1 Report
Good and clear data analysis. Also, good recommendations were given.
- The topic of this paper on “Lives saved in low- and middle-income countries by road safety” is somehow interesting. However,(1). The authors need to add some pictures to the paper (especially in the introduction section) to show the interesting side and get the reader's attention. (2). Draw some figures/flowcharts to show the workflow.
- The paper focus on the Lives saved in low- and middle-income countries by road safety is an essential topic in the introduction summarizing a lot of comparative literature but only include the city of the Bloomberg Philanthropies. It is better to add other live safety papers in different countries and measure the road safety value, such as the Jou, R. C., Chen, T. Y., (2015). The willingness to pay parties to traffic accidents for loss of productivity and consolation compensation. Accident Analysis & Prevention, 85, p.1-12.
- The author needs to explain some values more carefully, such as the cost-effectiveness ratio.
- Summarize and clarify the conclusion.
Reviewer 2 Report
See attached file for overall comment for improvement.

Reviewer 3 Report
Road safety is a crucial global issue. The paper tries to link changes in safety to actions of various organizations in this field. However, the attempt of linking the number of lives saved to actions is not underpinned by statistical evidences. Furthermore, the whole paper is biased very much towards the role of the authors' organization, which raises serious ethical concerns. It seems to be a commercial advertisment on a very serious issue.
Round 2
Reviewer 3 Report
I still believe that the paper is a commercial of the funding organizations.
Author Response
We appreciate the reviewer’s point of view and want to further assure them that we were an independent evaluation team who have no ties to the funding organisation.
Economic evaluations of the road safety laws and regulations funded by Bloomberg Philanthropies have been published previously: Miller TR, Levy DT, and Swedler DI. Lives saved by laws and regulations that resulted from the Bloomberg road safety pro-gram. Accident Analysis and Prevention. 2018;113:131-6. The reason for seeking an independent evaluation of its programs is to inform decision making about future funding of these programs.
The series of Bloomberg Philanthropy funded programs make a significant contribution to road safety and the findings of this publication will continue to inform future investment into road safety in low and middle income countries. Independent evaluations of these programs are critical to the efficient use of limited resources and provides a foundation for the implementation of evidence-based road safety initiatives.
The funding body has been acknowledged and the authors have declared no conflicts of interest. We hope the reviewer agrees on the importance of publishing this work in an open access journal.